# ConsisSR: Delving Deep into Consistency in Diffusion-based Image Super-Resolution

## Abstract

Real-world image super-resolution (Real-ISR) aims at restoring high-quality (HQ) images from low-quality (LQ) inputs corrupted by unknown and complex degradations. In particular, pretrained text-to-image (T2I) diffusion models provide strong generative priors to reconstruct credible and intricate details. However, T2I generation focuses on semantic consistency while Real-ISR emphasizes pixel-level reconstruction, which hinders existing methods from fully exploiting diffusion priors. To address this challenge, we introduce ConsisSR to handle both semantic and pixel-level consistency. Specifically, compared to coarse-grained text prompts, we exploit the more powerful CLIP image embedding and effectively leverage both modalities through our Hybrid Prompt Adapter (HPA) for semantic guidance. Secondly, we introduce Time-aware Latent Augmentation (TALA) to mitigate the inherent gap between T2I generation and Real-ISR consistency requirements. By randomly mixing LQ and HQ latent inputs, our model not only handle timestep-specific diffusion noise but also refine the accumulated latent representations. Last but not least, our GAN-Embedding strategy employs the pretrained Real-ESRGAN model to refine the diffusion start point. This accelerates the inference process to 10 steps while preserving sampling quality, in a training-free manner. Our method demonstrates state-of-the-art performance among both full-scale and accelerated models. The code will be made publicly available.

## 1 Introduction

With the increasing prevalence of image capturing devices in our daily lives, there emerges a growing need to capture clean, high-resolution images. Nevertheless, real-world images invariably contend with various degradation. To address this issue, real-world image super-resolution (Real-ISR) techniques adeptly reconstruct the high-quality (HQ) image from the low-quality (LQ) input.

Various methods (Dong et al., 2014; 2015; Kim et al., 2016; Lim et al., 2017; Zhang et al., 2018b; Dai et al., 2019; Niu et al., 2020) harness convolutional neural networks (CNN) to achieve remarkable performance, followed by transformer models (Liang et al., 2021; Zhang et al., 2022). Some others employ GANs' adversarial training to generate more photo-realistic images (Ledig et al., 2017; Wang et al., 2018). However, most of the above methods assume LQ inputs with basic bicubic down-sampling, limiting their efficacy in handling complex and unknown degradations encountered in Real-ISR. To tackle this problem, some methods manage to model the real-world degradations with complex degradation models, including degradation shuffle from BSRGAN (Zhang et al., 2021) and high-order degradation from Real-ESRGAN (Wang et al., 2021). Other GAN-based (Liang et al., 2022a;b; Chen et al., 2022) or GAN-prior (Menon et al., 2020; Pan et al., 2021; Chan et al., 2021) models also achieve impressive performance. While these methods enhance the perceptual quality of the reconstructed images, their generative capacity is limited and the training process in GANs is unstable, which may occasionally lead to unrealistic artifacts.

Recently, emerging diffusion models (DM) (Sohl-Dickstein et al., 2015; Ho et al., 2020; Song et al., 2020; Dhariwal & Nichol, 2021), particularly pretrained large-scale text-to-image (T2I) models including StableDiffusion (SD) (Rombach et al., 2022), have exhibited superior performance in image generation against GANs. Later with ControlNet (Zhang et al., 2023), various SD-based super-resolution (SDSR) methods can leverage pre-trained diffusion priors. Some of them (Wang et al., 2024a; Lin et al., 2023; Sun et al., 2023) neglect semantic embedding, resulting in sub-optimal per-

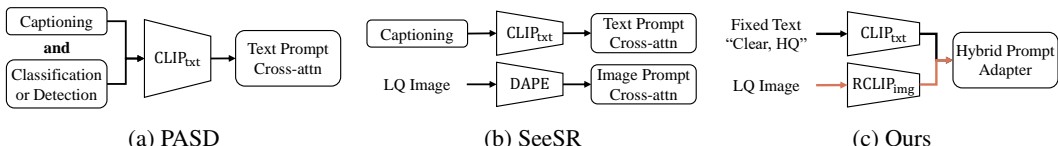

(a) PASD  (b) SeeSR  (c) Ours

Figure 1: Comparisons between existing semantic guidance for SDSR.

formance. As shown in Figure 1, some others (Sun et al., 2024; Yang et al., 2024) involve image captioning, which focuses on coarse-grained classification information while neglecting the color or texture details. SeeSR (Wu et al., 2024) integrates soft image prompts, but the misalignment between CLIP (Radford et al., 2021) and their DAPE results in additional attention layers. However, exploiting precise semantic guidance is inherently a complex task that requires extensive training. Given CLIP's joint text and image embedding, we introduce Hybrid Prompt Adapter (HPA) to adapt pretrained T2I cross-attention for joint image and text prompts to ensure semantic consistency.

Furthermore, T2I generation focuses on semantic consistency while Real-ISR emphasizes pixel-level reconstruction. We review the training process of DDPM (Ho et al., 2020) for these SDSR methods, which assumes all inputs consist of HQ latent data corrupted by Gaussian noise. We truncate the predicted $\hat{x}_{t\to0}$ at each step and decode them into images as shown in Figure 2.

It is evident that in the early sampling timesteps (t $\to 1000$), $\hat{x}_{t\to0}$ still appears overly smooth and noisy, which shows a clear discrepancy from the HQ distribution. As DDPM training assumes $\hat{x}_{t\to0}$ to be accurate HQ latents, the discrepancy between the predicted latent and the HR image would be accumulated and ultimately compromise the pixel-level consistency of the generated results during the sampling process. Therefore, we propose Time-aware Latent Augmentation (TALA) to ensure pixel-level consistency with the HQ target. By randomly selecting LQ or HQ latent and adding noise to it during training, our model not only learns to eliminate timestep-specific diffusion noise but also refines the early latent representations.

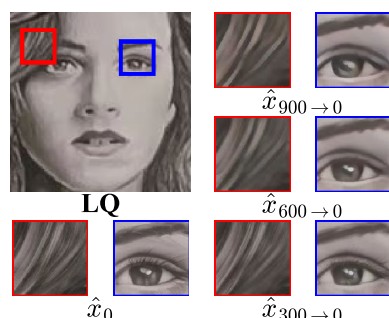

Figure 2: Visualization of the truncated outputs at different diffusion steps.

Furthermore, based on this observation, we also introduce the GAN-Embedding (GANEmb) strategy for inference acceleration. We dynamically embed the refined latent from Real-ESRGAN (Wang et al., 2021) into the start point of the diffusion sampling process and skip early diffusion steps. This allows our ConsisSR to reduce the diffusion process to at least 10 steps while preserving sampling quality, all in a training-free manner.

In this paper, we introduce ConsisSR to handle both semantic and pixel-level consistency for T2I diffusion prior. Specifically, we firstly propose HPA to incorporate the more powerful CLIP image embedding with text embedding to jointly enhance the semantic consistency. Then we review the questionable assumption in diffusion training, and put forward TALA to improve pixel-level consistency. TALA enables our model to not only remove timestep-specific diffusion noise but also refine the early latent representations. Lastly, our GANEmb strategy leverages pretrained Real-ESRGAN to skip early diffusion steps, accelerating the inference process while preserving the sampling quality. Our ConsisSR not only achieves SOTA results within the full-scale SDSR methods but also accelerated diffusion models. Our main contributions are as follows:

- Compared to coarse-grained text descriptions, we integrate the more powerful CLIP image embedding, which encapsulates additional color and texture details. Our Hybrid Prompt Adapter (HPA) simultaneously handles text and image prompts, adeptly leveraging T2I priors to enhance semantic consistency.

- We review the conventional SDSR training that assumes HQ latent data inputs. With Time-aware Latent Augmentation (TALA) between LQ and HQ latent, our ConsisSR is not only able to handle timestep-specific diffusion noise but also refine the latent representations predicted in early timesteps.

- Leveraging pre-trained Real-ESRGAN, our ConsisSR dynamically embeds refined latent and reduces the diffusion process to a minimum of 10 steps while preserving sampling quality, all in a training-free manner.

## 2 RELATED WORK

### 2.1 REAL-WORLD IMAGE SUPER-RESOLUTION

While deep learning-based image super-resolution has made notable advancements (Dong et al., 2015; Lim et al., 2017; Zhang et al., 2018b; Dai et al., 2019; Niu et al., 2020; Liang et al., 2021; Zhang et al., 2022), most of them rely on simple bicubic degradation, which restricts their effectiveness in handling complex and unknown degradations encountered in real-world scenarios. They also encounter challenges such as overly smoothed details when minimizing fidelity objectives. Real-world image super-resolution seeks to reconstruct photo-realistic image details by optimizing not just fidelity objectives but also perception objectives. Some works explore complex degradation models to approximate the real-world degradations, including degradation shuffle from BSR-GAN (Zhang et al., 2021) and high-order degradation from Real-ESRGAN (Wang et al., 2021). Further in (Menon et al., 2020; Pan et al., 2021; Chan et al., 2021), they leverage pretrained Style-GAN (Karras et al., 2019) as generative priors for Real-ISR. Even though they excel at robustly removing degradation in Real-ISR tasks, their limited generative capacity often hinders them from generating realistic details.

Emerging diffusion models Dhariwal & Nichol (2021); Rombach et al. (2022) exhibit a remarkable capability to generate high-quality images. When it comes to ISR tasks, several approach train their DMs from scratch on pixel space (Choi et al., 2021; Saharia et al., 2022). While the former necessitates tens to hundreds of diffusion steps, others (Xia et al., 2023; Chen et al., 2024) apply DM on compact latent space, but their generative ability is greatly restricted by the transformer backbone. Resshift (Yue et al., 2024) constructs a Markov chain to shift the residual between LQ and HQ images with only 15 steps. Additionally, SinSR (Wang et al., 2024b) advances this strategy to single-step with consistency preserving distillation. However, without strong diffusion priors, these methods still struggle to generate realistic and intricate textures.

### 2.2 SD-BASED SUPER-RESOLUTION

For pretrained DMs, ControlNet (Zhang et al., 2023) introduces an effective conditioning method, enabling broader applications. Based on this, various SD-based super-resolution (SDSR) methods have achieved unprecedented success. Some of them (Wang et al., 2024a; Lin et al., 2023; Sun et al., 2023) do not apply semantic embedding to guide the diffusion process, leading to sub-optimal performance. PASD (Yang et al., 2024) employ pretrained models including BLIP2 (Li et al., 2023) together with CLIP (Radford et al., 2021) text encoder to provide semantic guidance. Similarly, CoSeR (Sun et al., 2024) train a cognitive encoder to approximate the cascaded outputs of BLIP2 and CLIP. But they both involve the image captioning process, which focus on coarse-grained classification information while neglecting the color or texture details, as shown in Figure 4. SeeSR (Wu et al., 2024) fine-tunes RAM (Zhang et al., 2024) model and separately integrates text prompt and image prompts with additional cross-attention layers. However, given CLIP's proficiency in mapping text and image to a joint embedding space, the T2I priors from the SD model can also be applied to image prompts. Our decouple cross-attention harnesses our robust CLIP image encoder and exploits the domain alignment in CLIP embeddings, adapting our cross-attention for both text and image prompts.

Furthermore, all the aforementioned SDSR methods follow the training protocol from DDPM (Ho et al., 2020). We review the drawback in this training process, which assumes all inputs as HQ latent data with Gaussian noise. Through the proposed Time-aware Latent Augmentation (TALA), includes residuals between LQ and HQ latents we enable our ConsisSR to effectively eliminates diffusion noise while refining LQ latent to HQ latent.

## 3 METHOD

### 3.1 PRELIMINARY: DIFFUSION MODELS

Diffusion models are probabilistic models designed to generate data samples by gradually denoising a normally distributed variable $\mathbf{x}_T \sim \mathcal{N}(0, 1)$ in $T$ iteration steps. In each forward iteration, a Gaussian noise with variance $1 - \alpha_t$ is added to $\mathbf{x}_{t-1}$, and the overall forward process can be

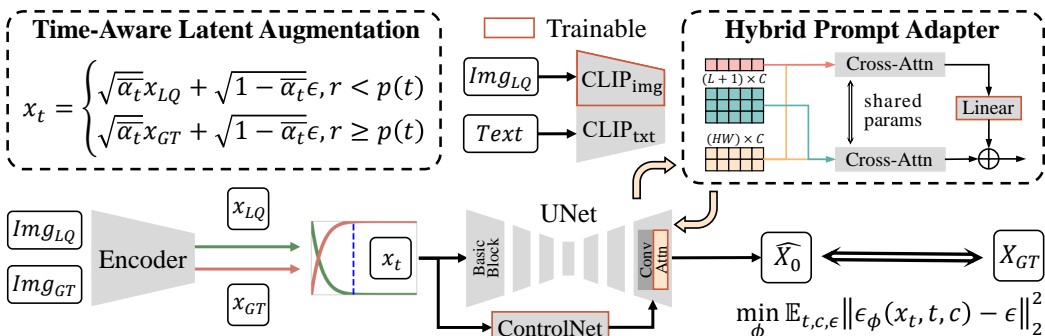

Figure 3: Overall training pipeline of our ConsisSR.

described as:

$$\mathbf{x}_t = \sqrt{\bar{\alpha}_t}\mathbf{x}_0 + \sqrt{1 - \bar{\alpha}_t}\epsilon, \tag{1}$$

where $\bar{\alpha}_t = \prod_{i=0}^{t} \alpha_i$ and $\epsilon \sim \mathcal{N}(0, 1)$. For the reverse process, diffusion models first sample a Gaussian noise $\mathbf{x}_T$ as the start point. Subsequently, conditioning on $c$, they iteratively estimate the added noise for each step $t$ through the denoising network $\epsilon_\phi$ until reaching the clean output $\hat{\mathbf{x}}_0$. The optimization objective is defined as follows:

$$\mathcal{L}_{DM} = \mathbb{E}_{\mathbf{x}, \epsilon \sim \mathcal{N}(0,1), t}\left[\|\epsilon - \epsilon_\phi(\mathbf{x}_t, t, c)\|_2^2\right]. \tag{2}$$

## 3.2 Overall Pipeline

As shown in Figure 3, we demonstrate our training augmentation on the left and the network architecture on the right. Firstly, we equip the diffusion UNet with ControlNet (Zhang et al., 2023) to manipulate the output using LQ images. Then we introduce the Hybrid Prompt Adapter (HPA), which leverages both CLIP's text and image embeddings as semantic guidance in a decoupled manner, thereby generating more realistic and credible texture. Furthermore, we put forward a Time-aware Latent Augmentation (TALA) for training during early timesteps, which enhances the pixel-level sampling consistency. We randomly select low-quality (LQ) and high-quality (HQ) latent inputs, enabling our model to not only remove timestep-specific diffusion noise but also predict the residual between LQ and HQ. Last but not least, by embedding refined latent from pretrained Real-ESRGAN (Wang et al., 2021), we are able to accelerate the diffusion process to 10 steps while maintaining sampling quality.

## 3.3 Hybrid Prompt Adapter

SD originates from the T2I generation task, hence original cross-attention is only tailored to text prompts. However, given that CLIP can inherently map images and text to a joint latent space, our intuition suggests that we can leverage the LQ image as conditioning. To better align the T2I generation task with Real-ISR, we propose the hybrid prompt adapter (HPA) to incorporate the input image prompt with the text prompt to enhance semantic consistency.

Although other SDSR methods also attempt to incorporate semantic guidance, including CoSeR (Sun et al., 2024), PASD (Yang et al., 2024), they both involve the image captioning process, which focus on coarse-grained classification information while neglecting the color or texture details. As shown in Figure 4, we present various text descriptions in a Venn diagram pattern. Following Radford et al. (2021), we calculate the cosine distances between the CLIP image embedding and different text embeddings to gauge the similarity between the input image and each text description. It can be observed that while BLIP2 offers a reasonable caption, it falls short in capturing the detailed information like color and accessory (white dog with red collar). The most precise and elaborate description exhibits the highest similarity, indicating the intricate details inherent in the image embedding itself. Therefore, we incorporate the more powerful CLIP image embedding with text prompt in our Hybrid Prompt Adapter, providing fine-grained semantic guidance for our denoising network, as depicted in Figure 5.

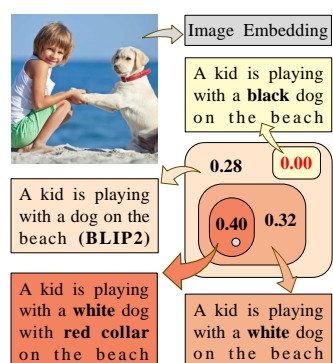

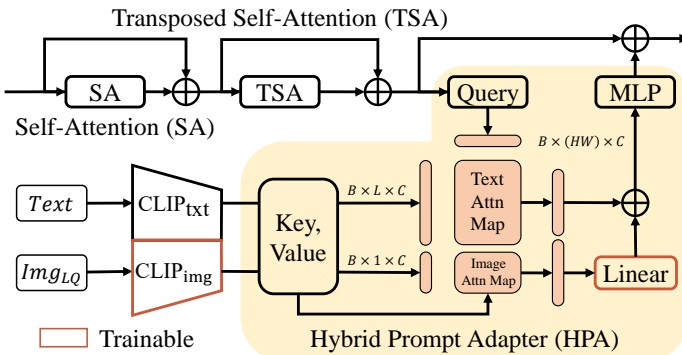

Figure 4: Similarity between image and text description.

Figure 5: Network architecture of our transformer block with Hybird Prompt Adapter (HPA).

Firstly, for the CLIP image encoder, we conduct a straightforward fine-tuning process to enhance its robustness to LQ images while preserving its alignment with the CLIP embedding domain.

$$\mathcal{L}_{CLIP} = \mathbb{E}_{\mathbf{x}_{HQ}, \mathbf{x}_{LQ}} \left[ \|\text{detach}(\text{CLIP}_{\text{img}}(\mathbf{x}_{HQ}) - \text{RCLIP}_{\text{img}}(\mathbf{x}_{LQ})\|_2^2 \right], \tag{3}$$

where $\text{CLIP}_{\text{img}}$ represents fixed original CLIP image encoder and $\text{RCLIP}_{\text{img}}$ represents our fine-tuned robust CLIP image encoder.

Then, the text prompt and low-quality (LQ) image are separately fed into the original CLIP text encoder and our robust CLIP image encoder, resulting in text and image prompt embeddings. Both prompt embeddings are subsequently transformed into Key and Value vectors using the original linear layers, which are then used to compute attention maps with Query vectors derived from the image latent. Akin to Ye et al. (2023), we employ a decoupled cross-attention, which involves independently processing the interactions between two pairs of Key and Value vectors with the Query vector. Differently, to preserve the SD prior as much as possible, we keep all projection layers fixed in our HPA for both denoising UNet and ControlNet. Correspondingly, the output of the additional image prompt branch is added to the original text prompt branch after passing through a trainable zero-initialized linear layer.

Apart from HPA, we also made adjustments to the self-attention layers. For the original self-attention, we incorporate learned absolute positional embeddings to better align with the image domain. Additionally, we introduce transposed self-attention inspired by the work in Zamir et al. (2022), which helps to further model global context. For the self-attention layers, we only train the added parameters in the denoising UNet.

## 3.4 Time-Aware Latent Augmentation

While diffusion models excel at approximating complex data distribution, ensuring the consistency of the generated results in this process is challenging. However, the SDSR task strongly emphasizes the pixel-level consistency of the generated images with targets, necessitating better handling of LQ latent representations, especially for the early steps which focus on structure refinement (Wang et al., 2024a). To achieve this goal, we propose Time-aware Latent Augmentation (TALA) to improve sampling consistency with the HQ target.

For T2I generation tasks, there are no specific requirements for detailed composition, including proportions and relative positions of different elements, but rather a focus on semantic similarity. However, for Real-ISR tasks, it is essential to prioritize structural consistency over blindly pursuing visual aesthetics. Although ControlNet (Zhang et al., 2023) introduces an effective conditioning method to provide pixel-level guidance, most SDSR methods still overlook certain drawbacks in the training of diffusion models.

To take a deeper insight, we take our ConsisSR as an example and truncate the predicted $\hat{x}_{t\to 0}$ at each step and decode them into images. We further quantified their reconstruction quality over timesteps in Figure 6. We can clearly observe that sampling consistency, represented by PSNR and SSIM, remarkably improves and remains stable until 40% (timestep 600). Subsequently, at the cost of consistency, the diffusion model steadily enhances texture details, leading to an improvement in

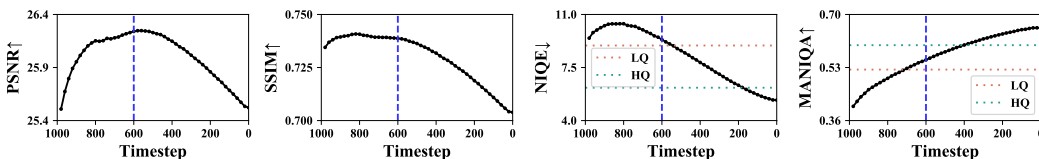

Figure 6: IQA metrics of the truncated outputs at different diffusion steps.

the generation quality indicated by NIQE and MANIQA. Moreover, in terms of generation quality, we can observe that the truncated results do not exhibit an advantage over the LQ input until timestep 600. This implies that in the early timesteps, the inputs to the denoising network lean more towards LQ latent data corrupted by Gaussian noise. This contrasts with the training process, which assumes that all inputs consist of HQ latent data.

Therefore, to enhance consistency while ensuring visual quality, we propose Time-aware Latent Augmentation which randomly selects LQ and HQ latent inputs, as shown in Algorithm 1. We devise a time-dependent probability function $p(t)$ where, for each timestep $t$, we probabilistically replace the input from HQ latent corrupted by Gaussian noise with the LQ one according to $p(t)$. Based on the afore-mentioned IQA over timesteps, we can divide the inference process into two phases: the former 40% focusing on consistency preservation and the latter 60% oriented towards visual aesthetics. Hence, our Time-aware Latent Augmentation (TALA) primarily operates during the first 40% timesteps. We set the starting point $p(1000)$ to be fixed at 1 and control $p(600)$ to be below 1%. Empirically, we adopt a power function $p(t) = (t/T)^\gamma$ and obtain $\gamma = 10$.

Following this augmentation, our supervision is no longer solely timestep-specific diffusion noise but also includes residuals between LQ and HQ latents. This indicates that our ConsisSR trained with TALA simultaneously removes diffusion noise and refines the latent representations predicted in the early steps, thereby enhancing sampling consistency.

| **Algorithm 1** Training Augmentation | **Algorithm 2** Inference Acceleration |
|---|---|
| **Require:** Paired training set $(\mathbf{X}_{LQ}, \mathbf{X}_{HQ})$ | **Require:** testing set $(\mathbf{X}_{LQ})$, timestep $t_{max}$ |
| 1: **while** not converged **do** | 1: **while** not converged **do** |
| 2:     sample $\mathbf{x}_{LQ}, \mathbf{x}_{HQ}$ from $(\mathbf{X}_{LQ}, \mathbf{X}_{HQ})$ | 2:     sample $\mathbf{x}_{LQ}$ from $(\mathbf{X}_{LQ})$ |
| 3:     sample $t \sim \mathcal{U}(\{1, ..., T\})$ | 3:     sample $\epsilon \sim \mathcal{N}(0, 1)$ |
| 4:     sample $\epsilon \sim \mathcal{N}(0, 1)$ | 4:     $\hat{\mathbf{x}}_0 = \text{GAN Embedding}(\mathbf{x}_{LQ})$ |
| 5:     sample $r \sim \mathcal{U}(0, 1)$ | 5:     $x_{t_{max}} = \sqrt{\bar{\alpha}_{t_{max}}}\hat{\mathbf{x}}_0 + \sqrt{1 - \bar{\alpha}_{t_{max}}}\epsilon$ |
| 6:     $\mathbf{x}_t = \begin{cases} \sqrt{\bar{\alpha}_t}\mathbf{x}_{LQ} + \sqrt{1-\bar{\alpha}_t}\epsilon, r < p(t) \\ \sqrt{\bar{\alpha}_t}\mathbf{x}_{HQ} + \sqrt{1-\bar{\alpha}_t}\epsilon, r \geq p(t) \end{cases}$ | 6:     **for** $t = t_{max}, ..., 1$ **do** |
| | 7:         sample $z \sim \mathcal{N}(0, 1)$ if $t > 1$ else $z = 0$ |
| 7:     $\hat{\epsilon} = \epsilon_\phi(\mathbf{x}_t, t, c)$ | 8:         $\mathbf{x}_{t-1} = \frac{1}{\sqrt{\alpha_t}}(\mathbf{x}_t - \frac{1-\alpha_t}{\sqrt{1-\bar{\alpha}_t}}\epsilon_\phi) + \sigma_t z$ |
| 8:     Update model with $\left\|\hat{\epsilon} - \frac{\sqrt{\bar{\alpha}_t}\mathbf{x}_{HQ} - \mathbf{x}_t}{\sqrt{1-\bar{\alpha}_t}}\right\|_2^2$ | 9:     **end for** |
| | 10:     return $\mathbf{x}_0$ |
| 9: **end while** | 11: **end while** |

### 3.5 GAN-EMBEDDING STRATEGY FOR INFERENCE ACCELERATION

Existing SDSR methods often require a large number of timesteps for inference. But unlike T2I generation tasks, SDSR involves LQ inputs and does not require starting from pure Gaussian noise. To address this issue, we propose GAN-Embedding (GANEmb) strategy for inference acceleration. We harness the pre-trained Real-ESRGAN (Wang et al., 2021) to pre-refine the LQ latent. Initiating the reverse process from this enhanced latent allows our model to skip early refinement steps and focus on detail generation. This compresses the inference process to a minimum of 10 steps while preserving sampling quality, all in a training-free manner.

Some methods attempt to reduce the inference steps by adjusting the noise schedule or sampling strategy Yue et al. (2024); Wang et al. (2024b); Sun et al. (2023). But they necessitate retraining the model, which sacrifices their ability to generate the most realistic and detailed textures. Instead, our GANEmb directly improves the inference process based on the trained SDSR model without additional training.

As discussed in the previous section, we empirically divide the inference process into the initial 40% sampling steps for consistency preservation and the last 60% sampling steps for detail generation.

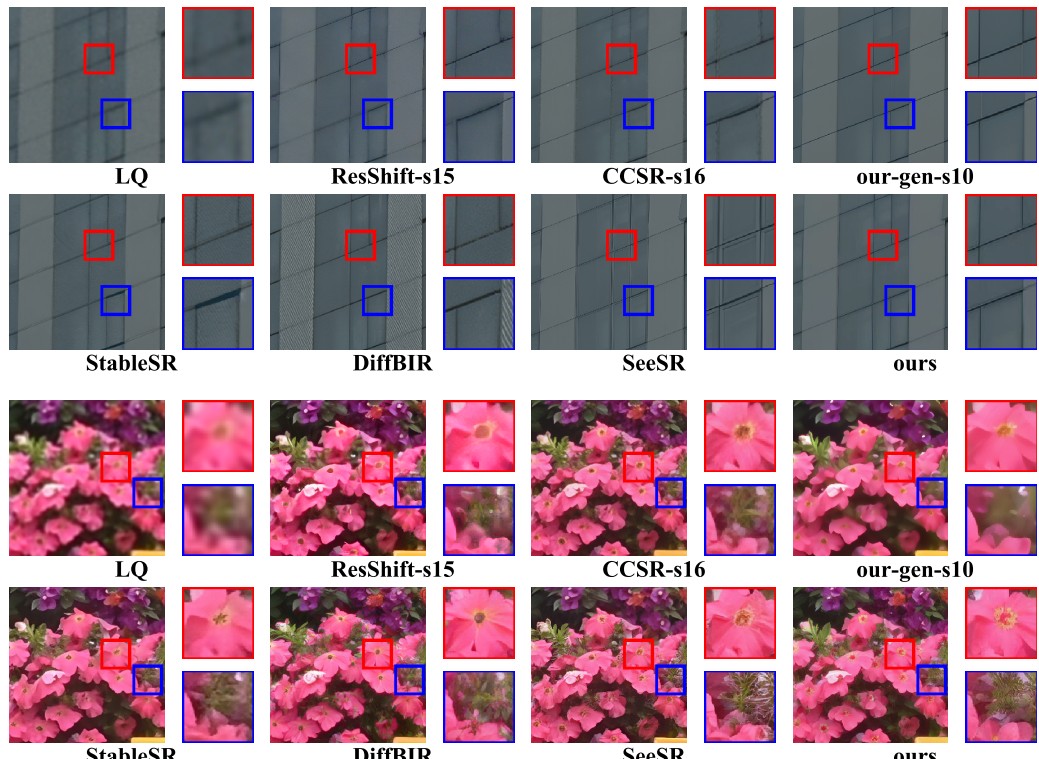

Figure 7: Visual comparisons among different Real-ISR methods. Please zoom in for a better view.

Although GAN methods exhibit limited generative capacity, they can effectively remove degradation and obtain a refined latent representation. This aligns with the early sampling timesteps in the diffusion model, to some extent. As shown in Algorithm 2, our GANEmb strategy leverages the pretrained Real-ESRGAN to skip early diffusion steps until $t_{max}$. We leverage the refined start point to reduce the model's reconstruction complexity, thereby accelerating the inference process while still maintaining the generative capabilities.

## 4 EXPERIMENTS

### 4.1 EXPERIMENTAL SETTINGS

**Datasets.** We train ConsisSR on the ImageNet dataset (Deng et al., 2009). We employ the degradation pipeline from Real-ESRGAN (Wang et al., 2021) to generate LQ-HQ training pairs. For testing datasets, we adopt the widely used DIV2K-Val (Agustsson & Timofte, 2017) as the synthetic dataset, along with RealSR Cai et al. (2019) and DrealSR Wei et al. (2020) as the real-world datasets.

**Evalution Metrics.** For quantitative evaluation of Real-ISR models, we first adopt traditional full-reference metrics, including PSNR, SSIM, and LPIPS (Zhang et al., 2018a) for consistency evaluation. Then for generation quality, we adopt the no-reference metrics, including NIQE (Mittal et al., 2012), MANIQA (Yang et al., 2022), MUSIQ (Ke et al., 2021) and CLIPIQA (Wang et al., 2023).

**Implementation details.** We first crop central patches from the ImageNet images and resize them to $224 \times 224$ for robust CLIP fine-tuning. The CLIP image encoder is trained with 2 NVIDIA L40s GPUs and the batch size is set to 8 per GPU. We adopt Adam as optimizer ($\beta_1 = 0.9$, $\beta_2 = 0.99$), and we train the model for 100K iterations with a learning rate fixed at $1 \times 10^{-5}$.

Then for SDSR training, the Stable Diffusion 2.1-base is used as the pretrained T2I model. We crop central patches of size $512 \times 512$ from ImageNet for training. The batch size is set to 2 per GPU, totaling 8 with 4 NVIDIA L40s GPUs. Our SDSR model is trained for 200K iterations with the Adam optimizer ($\beta_1 = 0.9$, $\beta_2 = 0.99$) and the learning rate is fixed at $5 \times 10^{-5}$. For 50-

Table 1: Qualitative Comparison with SOTA methods. The best results are highlighted in red and the second best results are highlighted in blue.

| Datasets | Methods | Full-reference IQA | | | No-reference IQA | | | | Rank(avg) |
|---|---|---|---|---|---|---|---|---|---|
| | | PSNR↑ | SSIM↑ | LPIPS↓ | NIQE↓ | MANIQA↑ | MUSIQ↑ | CLIPIQA↑ | |
| DrealSR | StableSR | 28.03 | 0.7536 | 0.3284 | 6.5239 | 0.5601 | 58.51 | 0.6356 | 4.00 |
| | DiffBIR | 26.71 | 0.6571 | 0.4557 | 6.3124 | 0.5930 | 61.07 | 0.6395 | 4.14 |
| | PASD | 27.36 | 0.7073 | 0.3760 | 5.5474 | 0.6169 | 64.87 | 0.6808 | 2.86 |
| | SeeSR | 28.17 | 0.7691 | 0.3189 | 6.3967 | 0.6042 | 64.93 | 0.6804 | 2.29 |
| | Ours | 28.47 | 0.7581 | 0.3463 | 6.3668 | 0.6224 | 65.28 | 0.6965 | 1.71 |
| RealSR | StableSR | 24.70 | 0.7085 | 0.3018 | 5.9122 | 0.6221 | 65.78 | 0.6178 | 4.00 |
| | DiffBIR | 24.75 | 0.6567 | 0.3636 | 5.5346 | 0.6246 | 64.98 | 0.6395 | 4.43 |
| | PASD | 25.21 | 0.6798 | 0.3380 | 5.4137 | 0.6487 | 68.75 | 0.6620 | 2.86 |
| | SeeSR | 25.18 | 0.7216 | 0.3009 | 5.4081 | 0.6442 | 69.77 | 0.6612 | 2.00 |
| | Ours | 25.51 | 0.7033 | 0.3223 | 5.2655 | 0.6552 | 69.48 | 0.6925 | 1.71 |
| DIV2K | StableSR | 23.26 | 0.5726 | 0.3113 | 4.7581 | 0.6192 | 65.92 | 0.6771 | 3.43 |
| | DiffBIR | 23.64 | 0.5647 | 0.3524 | 4.7042 | 0.6210 | 65.81 | 0.6704 | 3.86 |
| | PASD | 23.14 | 0.5505 | 0.3571 | 4.3617 | 0.6483 | 68.95 | 0.6788 | 3.14 |
| | SeeSR | 23.68 | 0.6043 | 0.3194 | 4.8102 | 0.6240 | 68.67 | 0.6936 | 2.57 |
| | Ours | 23.95 | 0.5896 | 0.3145 | 4.8323 | 0.6433 | 69.13 | 0.7153 | 2.00 |

step inference, we adopt the spaced DDPM sampling (Nichol & Dhariwal, 2021). For GANEmb inference, the sampling process only requires 10 timesteps.

## 4.2 COMPARISONS WITH SOTA METHODS

We compare our ConsisSR with diffusion-based SR methods, which are split into two groups: The first group, comprising full-scale models, primarily aims to generate the most realistic and detailed textures, which includes StableSR (Wang et al., 2024a), DiffBIR (Lin et al., 2023), PASD (Yang et al., 2024) and SeeSR (Wu et al., 2024). The second group focuses on accelerated diffusion models, achieving faster sampling speeds through adjustments in sampling strategy or model distillation, which include ResShift (Yue et al., 2024), SinSR (Wang et al., 2024b) and CCSR (Sun et al., 2023).

**Full-scale SDSR methods.** The quantitative results for these methods are presented in Table 1, and their average rank is listed in the last column. This comparison highlights the effectiveness of our model in handling Real-ISR tasks, showcasing its performance against other methods.

To emphasize the superior performance of our approach, we provide qualitative comparisons in Figure 7. For distinct and regular textures like glass curtain walls, our approach can produce sharper and well-aligned edges while preventing excessive artifact generation in smooth regions. For intricate and random textures like foliage and flowers, our method can also generate convincing and realistic texture details. These comparisons vividly illustrate that our method achieves superior SR results, marked by clearer textures and sharper edges.

**Accelerated diffusion methods.** To compare with these methods, we restrict the sampling steps of our ConsisSR to 10. By adjusting the start point $t_{max}$ in Algorithm 2, we introduce two models: $t_{max} = 600$ focusing more on reconstruction consistency, referred to as ours-rec, and $t_{max} = 1000$ emphasizing detail generation, known as ours-gen. Their results are demonstrated in Table 2 and Figure 7. We can find that our model continues to outperform other methods. Specifically, ours-rec and ours-gen each achieve exceptional performance on full-reference and no-reference metrics.

## 4.3 ABLATION STUDY

We further implement several variants of our method to demonstrate the effectiveness of each component in our model. Unless otherwise specified, our ablation models adhere to the same training settings as our ConsisSR model. And we report PSNR and LPIPS as full-reference metrics, along with NIQE and CLIPIQA as no-reference metrics on the RealSR dataset.

**Effectiveness of our transformer block.** Firstly, we make minor adjustments to the self-attention layers by integrating learned absolute positional embeddings into the original layers and introducing

Table 2: Qualitative Comparison with SOTA diffusion acceleration methods.The best results are in red and the second best results are in blue.

| Datasets | Methods | Full-reference IQA | | | No-reference IQA | | | | Rank(avg) |
| --- | --- | --- | --- | --- | --- | --- | --- | --- | --- |
| | | PSNR↑ | SSIM↑ | LPIPS↓ | NIQE↓ | MANIQA↑ | MUSIQ↑ | CLIPIQA↑ | |
| DrealSR | ResShift-s15 | 28.46 | 0.7673 | 0.4006 | 8.1249 | 0.4586 | 50.60 | 0.5342 | 4.33 |
| | SinSR-s1 | 28.36 | 0.7515 | 0.3665 | **6.9907** | 0.4884 | 55.33 | **0.6383** | 3.71 |
| | CCSR-s16 | **28.96** | 0.7710 | **0.2922** | **5.9100** | **0.5682** | 58.82 | 0.5467 | 2.29 |
| | Ours-rec-s10 | **29.46** | **0.8149** | **0.3132** | 8.5790 | 0.5484 | **59.89** | 0.6049 | 2.17 |
| | Ours-gen-s10 | 28.82 | **0.8020** | 0.3439 | 9.3679 | **0.5947** | **64.47** | **0.6837** | 2.50 |
| RealSR | ResShift-s15 | **26.31** | **0.7421** | 0.3460 | 7.2635 | 0.5285 | 58.43 | 0.5444 | 4.00 |
| | SinSR-s1 | **26.28** | 0.7347 | 0.3188 | **6.2872** | 0.5385 | 60.80 | **0.6122** | 3.00 |
| | CCSR-s16 | 26.24 | 0.7365 | **0.2559** | **5.7400** | 0.5974 | 63.64 | 0.5287 | 3.00 |
| | Ours-rec-s10 | 26.12 | **0.7495** | **0.3076** | 7.1198 | **0.6032** | **65.06** | 0.6063 | 2.00 |
| | Ours-gen-s10 | 25.48 | 0.7291 | 0.3399 | 7.6902 | **0.6393** | **68.73** | **0.6897** | 3.00 |
| DIV2K | ResShift-s15 | **24.65** | 0.6181 | 0.3349 | 6.8212 | 0.5454 | 61.09 | 0.6071 | 3.33 |
| | SinSR-s1 | 24.41 | 0.6018 | **0.3240** | **6.0159** | 0.5386 | 62.82 | 0.6471 | 3.43 |
| | CCSR-s16 | 24.46 | 0.6113 | **0.3045** | **4.6100** | **0.5912** | 62.78 | 0.5367 | 2.86 |
| | Ours-rec-s10 | **24.57** | **0.6310** | 0.3528 | 6.5367 | 0.5889 | **64.93** | **0.6679** | 2.50 |
| | Ours-gen-s10 | 24.34 | **0.6262** | 0.3751 | 7.5866 | **0.6011** | **66.60** | **0.7242** | 2.86 |

transposed self-attention to better capture global context. As depicted in the upper part of Table 3, these modifications led to a 0.4dB increase in PSNR with a slight drop in generation quality.

Then we focus on our proposed HPA module. We mainly compare with two other prompt extractor: the IP-Adapter (Ye et al., 2023) from image-to-image generation task and the DAPE soft prompt layers from SeeSR (Wu et al., 2024). The results are listed in the lower part of Table 3.

Table 3: Ablation studies on our transformer block.

| | Methods | Full-reference IQA | | No-reference IQA | |
| --- | --- | --- | --- | --- | --- |
| | | PSNR↑ | LPIPS↓ | NIQE↓ | CLIPIQA↑ |
| Attention Type | SA(baseline) | 24.76 | 0.3420 | 5.5183 | 0.6815 |
| | SA w/ PE | 24.91 | 0.3482 | **5.3348** | 0.6889 |
| | SA w/ PE+TSA | **25.16** | 0.3418 | 5.5665 | 0.6783 |
| Prompt Extractor | + IP-Adapter | 24.41 | 0.3560 | 5.4109 | **0.6947** |
| | + DAPE | 25.07 | **0.3219** | 5.4126 | 0.6878 |
| | + HPA | **25.50** | **0.3206** | **5.4083** | **0.6917** |

We can observe that IP-Adapter needs to make a significant sacrifice in consistency to achieve good generation results. This is primarily because its generation task only pays attention to semantic similarity rather than pixel-level consistency in Real-ISR. Regarding DAPE, its overall performance also falls short of our HPA.

The effective exploitation of precise semantic guidance is inherently a complex task that requires extensive training. Our HPA's superiority lies in leveraging the joint embedding space of CLIP to adapt the cross-attention prior from the image prompt, thereby enhancing the semantic consistency of the generated results.

**Effectiveness of our TALA training strategy.** Our TALA is primarily introduced to enhance the sampling consistency with HQ target. Therefore, we first demonstrate the full-reference metrics of our model with respect to timesteps with and without TALA, as shown in Figure 8. We repeat ten times for each parameter setting and calculate the mean value. It can be observed that during the early timesteps, our model trained with TALA achieved remarkable advantages in terms of PSNR, SSIM, and LPIPS.

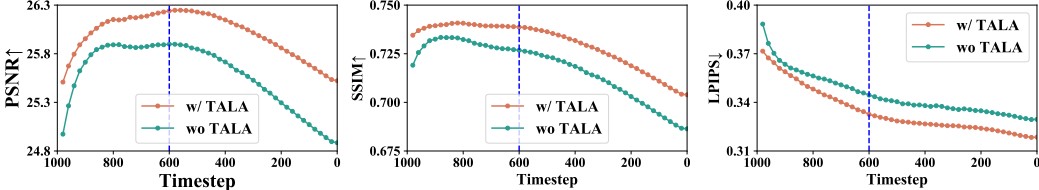

Figure 8: Full-reference IQA metrics over timesteps with and without TALA.

Furthermore, we conducted ablation experiments on the time-dependent probability function $p(t)$, as shown in Table 4. For the empirical parameter $\gamma$, we tested values of 5, 10, and 20, corresponding to strong, medium, and slight augmentations, respectively. It can be observed that even with slight augmentation ($\gamma = 20$) TALA significantly improves sampling consistency. The best trade-off

Table 4: Ablation studies on our Time-aware Latent Augmentation.

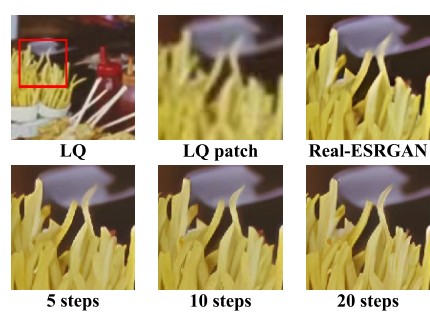

| Methods | Full-reference IQA | | No-reference IQA | |
|---|---|---|---|---|
| | PSNR↑ | LPIPS↓ | NIQE↓ | CLIPIQA↑ |
| DDPM | 24.77 | 0.3337 | 5.7709 | **0.6881** |
| TALA($\gamma = 5$) | 25.16 | **0.3197** | 5.7447 | 0.6624 |
| TALA($\gamma = 10$) | **25.50** | **0.3206** | **5.4083** | **0.6917** |
| TALA($\gamma = 20$) | **25.38** | 0.3280 | **5.4905** | 0.6811 |

between consistency and visual quality is achieved with $\gamma = 10$. However, when $\gamma$ decreases to 5, its generation capability rapidly deteriorates. This is primarily due to the excessive augmentation, which greatly compresses the detail enhancement process in the later timesteps.

**Effectiveness of our GANEmb inference strategy.** Regarding our inference strategy, there are two key parameters to determine: the start point $t_{max}$ for embedding GAN refined latent and the number of diffusion steps. By adjusting these parameters, we can dynamically achieve a trade-off between reconstruction consistency and detail generation without the need for additional training.

Firstly, we visualize the generated results for diffusion steps of 5, 10, and 20 in Figure 9. And it is evident that as the number of steps increases, the edges of each potato strip become sharper and more distinct. Furthermore, we quantify the trends in PSNR and CLIPIQA under different experimental settings as shown in Figure 10. And we also present various metrics for different start points with 10 diffusion steps.

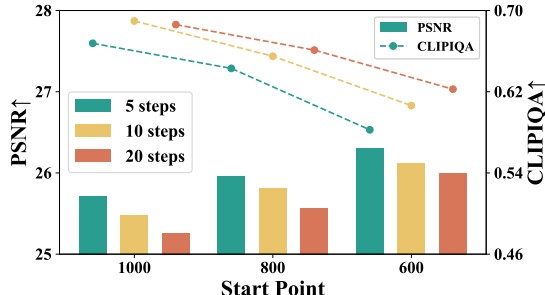

Figure 9: Generated results for different number of diffusion steps.

Figure 10: Ablation studies on GANEmb strategy, with PSNR in bar chart and CLIPIQA in line chart.

It can be clearly observed that as the start point decreases from 1000 to 600, the PSNR steadily increases while the CLIPIQA drops. This can be attributed to the higher noise intensity near 1000, resulting in a lower percentage of GAN refined latent and thereby freeing up more of the model's generative capacity. Similarly, under the same start point conditions, increasing the number of diffusion steps leads to lower PSNR and higher CLIPIQA. This aligns with our previous observations with the full-scale model, where more diffusion steps imply better texture detail capabilities.

## 5 CONCLUSION

Pretrained T2I diffusion models offer generative priors for Real-ISR task, yet they typically emphasize semantic consistency while neglecting pixel-level fidelity. To bridge the gap between T2I generation and Real-ISR tasks, we present ConsisSR, which adeptly exploit semantic and pixel-level consistency. Utilizing the powerful CLIP image embeddings, our Hybrid Prompt Adapter (HPA) seamlessly integrates both text and image modalities, providing semantic guidance for diffusion process. Furthermore, we introduce Time-aware Latent Augmentation (TALA) to improve pixel-level consistency in early timesteps. By randomly selecting LQ and HQ latent inputs, our model handles timestep-specific diffusion noise and refines latent states in the meantime. Additionally, our GAN-Embedding strategy, leveraging pretrained Real-ESRGAN, accelerates the diffusion process to just 10 steps without sacrificing quality. Our innovative approach achieves state-of-the-art results among both full-scale and accelerated diffusion models.

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
