# OpenReview forum: "ConsisSR: Delving Deep into Consistency in Diffusion-based Image Super-Resolution"
_ICLR.cc/2025/Conference — ICLR 2025 Conference Withdrawn Submission_

### Official Review · Reviewer_xyS8 · 2024-10-29

**Soundness:** 2
**Presentation:** 1
**Contribution:** 1
**Rating:** 3
**Confidence:** 5

**Summary:**

This paper focuses on utilizing pre-trained diffusion models to address real-world image super-resolution challenges. The authors argue that the discrepancy between the standard diffusion process, which emphasizes semantic consistency, and the Real-ISR problem, which necessitates pixel-level consistency, diminishes performance. To address this issue, they introduce a TALA strategy that randomly mixes LQ and HQ latent inputs. Additionally, they propose a HPA module to more effectively leverage the CLIP text/image embeddings, as well as a GAN-Embedding strategy to accelerate the sampling process

**Strengths:**

1. Their claim about the difference between T2I and Real-ISR seems interesting and benefiical.
2. The HPA seems to work.

**Weaknesses:**

1. The motivation behind TALA is somewhat unclear. I can not see the point why we use LQ to replace the HQ during training phase. How dose this strategy mitigate the discrepancy between semantic consistency and pixel-level consistency.
2. The values of GAN-embedding strategy appears to be limited. Firstly, this strategy is disconnected from the rest of the paper and appears to be incompatible with TALA (see questions). Secondly, it seems that this strategy has already been presented in previous works, such as CCDF. Thirdly, numerous one-step Real-SR diffusion models have recently been proposed, including SinSR, DFOSR, and OSEDiff. The value of a ten-step sampling strategy in the Real-SR field seems to be restricted.

[1] Chung, H., Sim, B., & Ye, J. (2021). Come-Closer-Diffuse-Faster: Accelerating Conditional Diffusion Models for Inverse Problems through Stochastic Contraction. 2022 IEEE/CVF Conference on Computer Vision and Pattern Recognition (CVPR), 12403-12412.
[2] Wang, Y., Yang, W., Chen, X., Wang, Y., Guo, L., Chau, L., Liu, Z., Qiao, Y., Kot, A.C., & Wen, B. (2023). SinSR: Diffusion-Based Image Super-Resolution in a Single Step. 2024 IEEE/CVF Conference on Computer Vision and Pattern Recognition (CVPR), 25796-25805.
[3] Wu, Rongyuan & Sun, Lingchen & Ma, Zhiyuan & Zhang, Lei. (2024). One-Step Effective Diffusion Network for Real-World Image Super-Resolution. 10.48550/arXiv.2406.08177.
[4] Zhang, A., Yue, Z., Pei, R., Ren, W., & Cao, X. (2024). Degradation-Guided One-Step Image Super-Resolution with Diffusion Priors. ArXiv, abs/2409.17058.

**Questions:**

1. The theoretical foundation of TALA appears to be confusing. For instance, in Algorithm 1, it is unclear why we can insert a LQ latent as input into the standard diffusion process, and why the loss function still involves the HQ latent when using the LQ latent as input.

2. The GAN-embedding strategy seems incompatible with TALA. Firstly, the GAN-embedding strategy is only mentioned in the inference phase and is not included in the TALA training strategy. Typically, we should aim to keep the training and inference processes as similar as possible. For example, why not train the models using the GAN-embedding strategy? Secondly, the GAN-embedding strategy requires the sampling process to skip a few steps in the early stage, while TALA aims to train a better diffusion for the early stage of the diffusion process. This suggests that the GAN-embedding strategy may undermine the effectiveness of the TALA strategy.

---

### Official Review · Reviewer_Sv8k · 2024-11-01

**Soundness:** 3
**Presentation:** 3
**Contribution:** 3
**Rating:** 5
**Confidence:** 4

**Summary:**

This paper proposes a real-world image super-resolution approach, aiming to align diffusion generative prior with super-resolution model learning. To achieve this, the authors propose Hybrid Prompt Adapter (HPA) for semantic guidance and Time-aware Latent Augmentation (TALA) to mitigate the inherent gap between T2I generation and Real-ISR consistency requirements. With all these improvements, the model achieves state-of-the-art performance on DrealSR, RealSR and DIV2K datasets comparing to previous milestone diffison-based models.

**Strengths:**

1.	The model achieves SOTA performances on several real-world image super-resolution datasets.
2.	The approach is easy to follow.
3.	The figures and visualizations are good and easy to understand.

**Weaknesses:**

1.	The proposed contributions seem incremental compared to the previous works, and all contributions are not for the same purpose. Some are (HPA) for the diffusion prior alignment and some are for sampling enhancement. More explanation is needed to conclude the contributions for consistency mentioned in the title.
2.	For TALA, how is the power function p(t) is determined and why? Is the \gamma selected empirically or by mathematical proof? Is the function of p(t) the best choice? Some more ablation studies may be needed at least to show the effectiveness of this function.
3.	The finetuning of of RCLIP with CLIP loss mainly helps the tuned CLIP model to be aware of the degradation and low-quality image inputs in feature space, which is somehow incremental since the previous works [1][2] have had similar network designs.
4.	Note that the ConsisSR uses SD-2.1 as the base model. This may be unfair when comparing to other models like SeeSR [1], which adopts SD-2 as the base model. More ablations are needed to eliminate the gain brought by the base model.

[1] SeeSR: Towards Semantics-Aware Real-World Image Super-Resolution (CVPR 2023)
[2] TPGSR: Text Prior Guided Scene Text Image Super-resolution (TIP 2023)

**Questions:**

How is the power function p(t) is determined and why? Is the \gamma selected empirically or by mathematical proof? Is the function of p(t) the best choice?

---

### Official Review · Reviewer_TjzG · 2024-11-02

**Soundness:** 3
**Presentation:** 3
**Contribution:** 3
**Rating:** 6
**Confidence:** 4

**Summary:**

This paper introduces a real-world image super-resolution method aimed to improve semantic and pixel-level consistency of generated results. First, in order to improve semantic consistency, the paper proposes to incorporate CLIP image embeddings in the SDSR framework. To that end, they propose to replace the cross-attention layers in the denoising UNet with a new block, called Hybrid Prompt Adapter, which processes the CLIP text and image embeddings with a decoupled cross-attention strategy. Then, to improve pixel-level consistency of the generated results, the paper introduces the Time-aware Latent Augmentation strategy. The main principle of this augmentation strategy is to randomly select between low quality and high quality noised latents during the denoiser network training on early diffusion steps, instead of only training on high quality noisy latent inputs, as done in previous works. As an additional contribution, the paper explores reducing the number of diffusion timesteps during sampling by leveraging the result of pre-trained Real-ESRGAN. They skip early diffusion steps, and start the sampling process from the Real-ESRGAN result’s embedded latent representation, which allows to achieve good quality with less number of sampling steps. Extensive experimental results show the effectiveness of the proposed contributions.

**Strengths:**

1. The proposed method has a clear motivation and quantitatively achieves better overall results in comparison to existing state-of-the-art methods.
2. The paper performs extensive ablation studies to verify the effectiveness of each change.
3. The presentation quality of the paper is good. The proposed method is clearly understandable.

**Weaknesses:**

1. The paper has a very limited number of qualitative comparison results. Authors should consider adding supplementary material with more results.
2. In Table 2 the paper lacks a comparison with [1], which is also an accelerated diffusion model for real-world image super-resolution.
3. The paper lacks a user study.

References

[1] Wu, Rongyuan, Lingchen Sun, Zhiyuan Ma, and Lei Zhang. "One-Step Effective Diffusion Network for Real-World Image Super-Resolution." arXiv preprint arXiv:2406.08177 (2024).

**Questions:**

1. Shouldn’t in Algorithm 1 on line 8 be: $\left\| \hat{\epsilon} - \frac{x_t - \sqrt{\bar{\alpha}_t} x_{\text{HQ}}}{\sqrt{1 - \bar{\alpha}_t}} \right\|^2_2$ ?

2. Fig. 4 is not very clear. What is 0.0 text in red color? Does it mean that the cosine similarity between the corresponding text embedding and image embedding is 0 (how is it possible)?

---

### Official Review · Reviewer_YvVA · 2024-11-03

**Soundness:** 2
**Presentation:** 2
**Contribution:** 2
**Rating:** 5
**Confidence:** 4

**Summary:**

The paper addresses the consistency issue in real-world super-resolution by proposing the Hybrid Prompt Adapter (HPA) and Time-aware Latent Augmentation (TALA). The former enhances the model's semantic consistency using CLIP image and text embeddings, while the latter employs a dynamic strategy to select LQ and HQ images to improve pixel-level consistency. Additionally, the authors propose a training-free accelerated sampling strategy that uses GAN embeddings as the starting point for SD, effectively reducing the sampling steps to 10.

**Strengths:**

The network diagrams in the paper are clear and easy to understand.

**Weaknesses:**

1. The definition and explanation of consistency are unclear. The authors do not clearly specify what semantic consistency is or what metrics can reflect this consistency. The authors claim that HPA provides more fine-grained semantic guidance, suggesting that HPA can help SD generate more accurate content and enhance its generative capability. However, why does Table 3 show that HPA improves PSNR and LPIPS? Furthermore, the authors state that TALA is meant to enhance pixel-level consistency, with fidelity-related metrics reflecting this consistency, yet Table 1 shows no significant advantage in PSNR, SSIM, and LPIPS for the model.

2. The proposed accelerated sampling strategy is not novel, as similar operations have already been used in DiffBIR.

3. The explanation of the motivation behind TALA is vague. The paper lacks a persuasive  explanation and experimental evidence to support why this approach can improve pixel-level consistency. The explanation of this module in lines 290-294 is quite abstract and difficult to understand.

4. In Equation 3, the parentheses are mismatched, and the detach operation does not have a corresponding right parenthesis.

**Questions:**

The authors need to clearly define what semantic consistency and pixel-level consistency are, as well as which metrics can reflect this consistency. Additionally, they should ensure that the experimental results validate the effectiveness of each module.

---

### Note · Authors · 2024-11-15

I have read and agree with the venue's withdrawal policy on behalf of myself and my co-authors.